# GLACIAL: Granger and Learning-based Causality Analysis for Longitudinal Studies

## Abstract

The Granger framework is useful for discovering causal relations in time-varying signals. Granger causality (GC) tools are mostly developed for densely sampled timeseries data. A substantially different setting, particularly common in population health applications, is the longitudinal study design, where *multiple* individuals are followed and sparsely observed over time. Longitudinal studies commonly track many variables, which are likely governed by nonlinear dynamics that might have individual-specific idiosyncrasies and exhibit both direct and indirect causes. Furthermore, real-world longitudinal data often suffer from widespread missingness. GC methods are not well-suited to handle these issues. In this paper, we propose an approach named GLACIAL (Granger and LeArning-based CausalIty Analysis for Longitudinal studies) to fill this methodological gap by marrying GC with a multi-task neural model. GLACIAL treats individuals as independent samples and uses the model's average prediction accuracy on hold-out individuals to probe causal links. Input feature dropout and model interpolation are used to efficiently learn nonlinear dynamic relationships between a large number of variables and to handle missing values respectively. Our experiments on synthetic and real data show GLACIAL outperforming competitive baselines and confirm its utility.

## 1 Introduction

Granger causality (GC) (Granger, 1969) is a versatile and popular framework that exploits "the arrow of time" to detect causal relations in timeseries data (Roebroeck et al., 2005; Zhang et al., 2011). In GC, we test whether past values of one time series help in predicting future values, which allows us to infer causal relationships. Despite its popularity, current implementations of GC are only well-suited for densely and uniformly sampled timeseries data from one system at a time. They are not designed for longitudinal studies, involving multiple systems (e.g., individuals). Although one could infer a causal graph for each individual and aggregate the graphs across individuals, this approach is untenable in many longitudinal studies where each individual only has a few observations, making the inference of each causal graph inaccurate or impossible.

Constraint-based methods such as PC or FCI (Spirtes et al., 2000), which rely on independent samples and conditional independence tests, are also commonly used for causal discovery. These methods would use one observation per individual and thus is not designed to detect causal relations reflected in temporal dynamics. We believe there is a lack of methods for causal discovery in longitudinal studies that consist of multiple individuals with sparse observations.

There are other issues that make causal discovery in longitudinal studies challenging. Longitudinal studies usually track multiple variables and the relationships between these variables may be nonlinear, which can be hard to detect. For instance, using linear GC to infer nonlinear relationships can be fast but may produce wrong results (Li et al., 2018). On the other hand, nonlinear GC methods (e.g. those based on non-parametric methods (Su & White, 2007; Marinazzo et al., 2008)) do not scale to large number of variables (Eichler, 2012). Similarly, existing GC tests that use neural networks to infer nonlinear dynamics (Tank et al., 2021; Nauta et al., 2019; Khanna & Tan, 2020) also face scalability issues. Furthermore, prior GC methods are, to the best of our knowledge, all association-based. That is, they test for causal relationships via interrogating fit

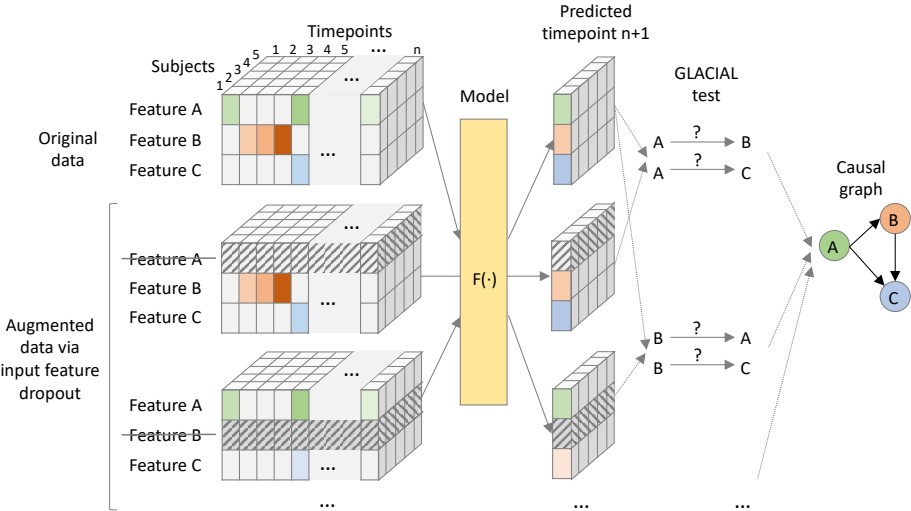

Figure 1: *Overview of the proposed GLACIAL approach for longitudinal studies.*

(learned) model weights. For example, in the linear GC approaches, this is achieved by testing the statistical significance of model coefficients. As the detection power of association-based GC (Granger, 1969; Lütkepohl, 2005) diminishes with increasing number of variables Sugihara et al. (2012); Runge et al. (2019b), it may fail to detect the weak coupling between a node and its parents, in particular when there are a lot of variables and limited data (Runge et al., 2019a; Yuan & Shou, 2021). Another challenge of real-world longitudinal studies is missing data. While there is no consensus about what to do about missing values (Glymour et al., 2019), several works (Strobl et al., 2018; Tu et al., 2019) have tried to address this issue for cross-sectional data. Yet, as far as we know, missingness is under-explored in longitudinal studies, particularly in the context of causal discovery. Finally, GC, in its original form, does not differentiate between direct and indirect causes (Yuan & Shou, 2021). Although, in theory, infinite history (observations) could shield off indirect causes from being detected as edges in the output causal graph, when the number of observations per individual is small, false positives due to indirect causes is a common practical problem.

In this work, we propose GLACIAL (Figure 1), which stands for a "Granger and LeArning-based CausalIty Analysis for Longitudinal studies." GLACIAL combines GC with a practical machine-learning based approach to test for causal relations between multiple variables in a longitudinal study. GLACIAL extends GC to longitudinal studies by treating each individual's trajectory as an independent sample, governed by a shared causal mechanism that is reflected in the temporal dynamics. This treatment is similar to prior works where individuals are assumed to be independent samples in longitudinal data analysis (Hernan & Robins, 2020). By applying a standard train-test setup with hold-out individuals, GLACIAL can test for effects of causal relations in expectation. Critically, GLACIAL infers causal relationships based on interrogating predictive accuracy and not a direct analysis of model weights, which is common in existing association-based GC methods. GLACIAL employs a single multi-task neural model, trained with input feature drop-out, to learn nonlinear relationships between all variables in time-varying data. The model also handles missing values using model interpolation. Thus, although neural networks have been used in the past for causal discovery, GLACIAL efficiently tests for causal relations of a large set of variables in real-world data where timepoints may be sampled irregularly and may contain missing values. Furthermore, GLACIAL includes post-processing heuristics to account for indirect causes and resolve the directionality of detected ambiguous associations. Extensive experiments on synthetic and real longitudinal data show that GLACIAL can infer relationships accurately even in challenging real-world scenarios with sparse observations, a large number of variables and direct causes, and a large degree of missing data. Although a specific model was used in our experiments, GLACIAL is model-agnostic.

## 2 Background

Most existing causal discovery (CD) methods are not intended for the longitudinal study design, where multiple individuals are sparsely observed at different timepoints. CD methods designed for timeseries data or independent samples are often used in the longitudinal setting despite potential poor performance.

**Causal Discovery**  CD methods intended for cross-sectional studies are ill-suited for longitudinal studies. They often fall under: constraint-based search (e.g. PC and FCI (Spirtes et al., 2000)), score-based search (e.g. GES (Chickering, 2002)), functional causal models (FCMs, e.g. LiNGAM (Shimizu et al., 2006), ANM (Hoyer et al., 2008; Zhang & Hyvärinen, 2009a), and PNL (Zhang & Chan, 2006; Zhang & Hyvärinen, 2009b)), or continuous optimization (e.g. NOTEARS (Zheng et al., 2018)). Search methods can scale well if causal relations are linear (Kalisch & Bühlman, 2007; Ramsey et al., 2017) although their output may not be informative enough (e.g. contains bidirectional edges). In contrast, by making strong assumptions about the functional form of the causal process, FCM can better identify the causal direction (Hyvärinen & Pajunen, 1999; Zhang et al., 2015), although FCM methods usually do not scale well (Glymour et al., 2019). Besides, if the assumed FCM is too restrictive to be able to approximate the true data generating process, the results may be misleading.

There are also various CD methods for timeseries such as ANLTSM (Chu et al., 2008), PCMCI(+) (Runge et al., 2019b; Runge, 2020) (based on PC), tsFCI (Entner & Hoyer, 2010) and SVAR-(G)FCI (Malinsky & Spirtes, 2018; 2019) (based on FCI), VAR-LiNGAM (Hyvärinen et al., 2010) (based on LiNGAM), TiMINo (Peters et al., 2013) (based on FCM), or DYNOTEARS (Pamfil et al., 2020) (based on NOTEARS). These methods take in consecutive blocks of observations and output a Full Time Graph (Peters et al., 2017), which contain not only the variables in the system but also their temporally-lagged versions. Although methods for timeseries may be better than cross-sectional ones, they are still not ideal for longitudinal data where sparse observations with potentially missing values come from more than one individual.

**Granger Causality**  GC (Granger, 1969; 1980) checks for dependence between variables' timeseries, after accounting for other available information. Temporal dependence is thus linked to causation by the "Common Cause Principle": two dependent variables are causally related (one causes the other, or both share a common cause) (Peters et al., 2017). Checking pairwise dependence in GC can be efficient, but often yields false positives because other variables in the system are not accounted for. In contrast, multivariate GC can account for common causes and therefore is more accurate but also more computationally demanding (Eichler, 2007; 2012). In practice, multivariate GC may be infeasible for a large set of variables and more efficient approaches (Basu et al., 2015; Huang & Kleinberg, 2015) were developed to deal with this challenge. Recently, more general GC tests based on neural networks (Tank et al., 2021; Nauta et al., 2019; Khanna & Tan, 2020) have been proposed which outperform vector auto regressive (VAR) linear GC (Glymour et al., 2019). Scaling these neural-network based GC methods to handle a large number of variables is still a concern.

**Missing data**  For cross-sectional studies, missing values can be imputed, which may result in data contradicting the causal processes. Alternatively, observations with missing values can be removed (list-wise deletion), which can lead to the omission of vast amounts of valuable datapoints. Test-wise Deletion (Strobl et al., 2018) (TDPC) is more data-efficient than list-wise deletion but may produce spurious edges when missingness is not completely at random (Tu et al., 2019). MVPC (Tu et al., 2019) corrects TDPC's output to account for different missingness scenarios. To our knowledge, no existing method addresses missingness for CD in longitudinal studies.

## 3 Method

Both cross-sectional CD methods (multiple individual, single timepoint data) and timeseries CD methods (single individual, multiple timepoints data) are ill-suited for longitudinal studies (multiple individuals, multiple timepoints data). Besides, prior methods often assume timeseries are infinitely long (i.e. unlimited history), regularly sampled, and without missing values. Thus, they may not work for real-world datasets when observation history per individual is limited, irregular, and riddled with missing values. Section 3.2

shows how GLACIAL handles longitudinal data. Section 3.3 shows how GLACIAL deals with irregularly sampled timepoints containing missing values. Section 3.4 shows GLACIAL's post-processing strategies to account for limited history of observed timeseries.

Causal discovery is impossible without assumptions. GLACIAL assumes causal faithfulness, no hidden confounder, acyclicity (DAG) and no instantaneous effects (the first three assumptions are standard in CD literature, c.f. Pearl (2009)). GLACIAL does not assume stationarity unlike linear GC.

### 3.1 Longitudinal Study Set-up

In a longitudinal study, there are *multiple* individuals who are sparsely observed for a limited number of times. Let $\mathbf{X}_t$ and $\mathbf{Y}_t$ be time-varying variables indexed with positive integer $t \in \{0, \ldots T-1\} = [T]$. We use super-script notation to indicate history: $\mathbf{X}^t = \{\mathbf{X}_0, \ldots, \mathbf{X}_{t-1}\}$. $\mathbf{\Omega}^t = \mathbf{X}^t \cup \mathbf{Y}^t \cup \ldots$ is the union of histories of all variables. The data from individual $i$ with $T_i$ observations ($T_i \leq T$) is $\mathbf{\Omega}^{T_i}$. The whole longitudinal dataset is $\{\mathbf{\Omega}^{T_i}; \ i \in 1, \ldots, N\}$. The number of observations, $T_i$, is usually less than 10 (sparse) while the number of individuals, $N$, is usually less than 10000. The $\mathbf{\Omega}^{T_i}$ matrices may contain missing values.

### 3.2 Granger Causality Formulation

A popular GC test is based on comparing the mean squared error (MSE) achieved by two predictors (Granger, 1980). In the GC MSE formulation, we conclude that "$Y$ causes $X$" if:

$$\delta_t(X|Y) = \mathsf{MSE}(\mathbf{X}_t, \mathbb{E}[\mathbf{X}_t | \mathbf{\Omega}^t \setminus \mathbf{Y}^t]) - \mathsf{MSE}(\mathbf{X}_t, \mathbb{E}[\mathbf{X}_t | \mathbf{\Omega}^t]) > 0, \forall t \in [T] \tag{1}$$

where $\mathbb{E}$ denotes (conditional) expectations. Equation 1 simply calculates the MSE difference between two optimal (in an MSE sense) predictors of $X$ (also see (Granger, 1980) and Appendix B). The first predictor (i.e. $\mathbb{E}[\mathbf{X}_t | \mathbf{\Omega}^t \setminus \mathbf{Y}^t]$) is not given information about $Y$. The second predictor (i.e. $\mathbb{E}[\mathbf{X}_t | \mathbf{\Omega}^t]$) is given all past information, including about $Y$. Since $\delta_t(X|Y) > 0, \forall t$, Equation 1 can be adapted for longitudinal data as:

$$\Delta\mathsf{MSE}(X|Y) = \mathbb{E}_i\Big[\frac{1}{T_i} \sum_{t=0}^{T_i-1} \delta_t(X|Y)\Big] > 0 \tag{2}$$

Relying on the assumption that statistical dependence implies a causal link (Reichenbach, 1956), when past values of $Y$ predict future values of $X$ (dependence): (1) $X$ causes $Y$ OR (2) $Y$ causes $X$ OR (3) $X$ and $Y$ have a common cause. The fact that causes occur before effects in time rules out (1) while the no hidden confounder assumption rules out (3). Hence, a positive test implies that "$Y$ causes $X$". Section 3.3 details how this test can be done in practice when the optimal predictors are not given. We are particularly interested in the setting with multiple observed independent individual trajectories.

### 3.3 Choice of Predictor

We can approximate the MSE-optimal predictors with neural networks $F$ and $G$.

$$\delta_t(X|Y; F, G) = \mathsf{MSE}(\mathbf{X}_t, F(\mathbf{X}_t; \mathbf{\Omega}^t \setminus \mathbf{Y}^t)) - \mathsf{MSE}(\mathbf{X}_t, G(\mathbf{X}_t; \mathbf{\Omega}^t)) \tag{3}$$

$$\Delta\mathsf{MSE}(X|Y; F, G) = \mathbb{E}_i\Big[\frac{1}{T_i} \sum_{t=1}^{T_i} \delta_t(X|Y; F, G)\Big] \tag{4}$$

To calculate $\delta_t(X|Y; F, G)$, we first have to train the neural networks using a training set. Once trained, the neural networks can be used to calculate $\Delta\mathsf{MSE}(X|Y; F, G)$ on hold-out test individuals. Thus, the predictors' performance depends on the training data, optimization, network initialization, and other implementation details. Even with the best optimizer and initialization procedure, a bad training-test split could, for instance, result in a sub-optimal model and consequently false causal link estimates. For more robust causal discovery, in GLACIAL, we repeat the estimation of $\Delta\mathsf{MSE}(X|Y; F, G)$ multiple times using different random splits of data and test that $\Delta\mathsf{MSE}$ is positive on average using a statistical test.

We use a *single* recurrent neural network (RNN) (Graves et al., 2009) in place of all predictors. The RNN is trained to predict the next step values of all the variables, $\boldsymbol{\Omega}_t$, given all available past values, $\boldsymbol{\Omega}^t$. We adopt the RNN model from Nguyen et al. (2020) since it implements model interpolation to handle missing values. In particular, if there are missing values at time $t$, they can be replaced by the RNN prediction, $\widehat{\boldsymbol{\Omega}}_t$ (model interpolation). This timepoint with interpolated values is then concatenated with $\boldsymbol{\Omega}^t$ to form $\boldsymbol{\Omega}^{t+1}$ which is subsequently used to predict values at time $t+1$. Missing values are ignored when calculating training loss and estimating $\delta_t(X|Y; F, G)$ on hold-out individuals (Equation 3). Training follows (Nguyen et al., 2020) so that even data containing missing values can be used for RNN training. Note that the choice of neural network model is not critical. Any model that forecasts future values from past values and implements model interpolation should work in GLACIAL.

**Input Feature Dropout**  Training separate neural networks to compute $\Delta\mathsf{MSE}(X|Y; F, G)$ for each variable pair would create a substantial burden for applying this approach to systems with large number of variables. This is because the number of networks required would be proportional to the number of variables squared. Instead, we propose to train a single multi-task (i.e., multi-output) RNN, $F(\cdot; \theta)$, to approximate $\mathbb{E}[\mathbf{X}_t|\boldsymbol{\Omega}^t \setminus \mathbf{Y}^t]$ and $\mathbb{E}[\mathbf{X}_t|\boldsymbol{\Omega}^t]$, for all predicted variables $\mathbf{X}_t$. The RNN acts as the former when $Y$ is masked out of the input vector and acts as the latter when the input is complete. To obtain a model that can produce accurate predictions under these scenarios, during training, we augment each mini-batch by dropping out individual variables from the input features.

---

**Algorithm 1:** GLACIAL

**Input:** Data splits $(D_1^{train}, D_1^{test}), \ldots, (D_n^{train}, D_n^{test})$
**Output:** Causal graph $G$
  // Step 1:  Association detection using the GC MSE test
**1 for** *each data split $D_i$* **do**
**2**     Fit RNN model $F_i$ using $D_i^{train}$
**3**     **for** *each variable pair $(u, v)$* **do**
**4**       Calculate $\Delta\mathsf{MSE}[u, v, i]$ using $F_i$ and $D_i^{test}$;

**5 for** *each variable pair $(u, v)$* **do**
**6**     t-statistic, p-value = t-test($\Delta\mathsf{MSE}[u, v, *]$)
**7**     **if** *p-value $<$ threshold* **then**  Add $u{\to}v$ to $G$;     S[u, v] = t-statistic;

  // Step 2:  Orient bidirectional edges
**8 for** *each bidirectional pair $u{\to}v$ and $v{\to}u$ in $G$* **do**
**9**     **if** $S[u, v] < S[v, u]$ **then**  Remove $u{\to}v$ from $G$;            `// `$v{\to}u$` has stronger effect`
**10**    **else**  Remove $v{\to}u$ from $G$;                     `// `$u{\to}v$` has stronger effect`

  // Step 3:  Prune indirect causes
**11 for** *each $u{\to}v$ in $G$* **do**
**12**    **for** *each path $p = (u{=}w_0, w_1, \ldots, w_k{=}v)$* **do**
**13**       **if** $S[u, v] < S[w_j, w_{j+1}]$ $\forall j \in \{0, \ldots, k{-}1\}$ **then**  Remove $u{\to}v$;    break;
**14**    **for** *each path $p = (v{=}w_0, w_1, \ldots, w_k{=}u)$* **do**
**15**       **if** $S[u, v] < S[w_j, w_{j+1}]$ $\forall j \in \{0, \ldots, k{-}1\}$ **then**  Remove $u{\to}v$;    break;

---

**Implementation Details**  The same settings of GLACIAL are used in all experiments. We used repeated 5-fold cross-validation to split a dataset into training, validation, and test sets with a 3:1:1 ratio. The RNN is trained to minimize next-step prediction error using Adam (Kingma & Ba, 2014), L2 loss, and a learning rate of 3E-4. The RNN has one hidden layer of size 256. Training was done on a NVIDIA TITAN Xp GPU. The validation set is used for early stopping. Cross-validation is repeated 4 times, resulting in 20 different splits of data. We find 4 repetitions to strike a good balance between robustness and speed. Running more repetitions might slightly improve the results when missingness is severe but at a higher computational cost (see Appendix E). We perform a t-test on the $\Delta\mathsf{MSE}$ statistic and use the significance level threshold of 0.05.

### 3.4 Post-Processing

GC assumes history of the timeseries is infinite. When observations are finite as in real-world longitudinal studies, GC may draw wrong conclusions. E.g., consider following deterministic system:

$$\mathbf{Y}_t = a\mathbf{Y}_{t-1} + b\mathbf{Y}_{t-2}$$
$$\mathbf{X}_t = c\mathbf{Y}_{t-2}.$$

In this system, $Y$ causes $X$ since manipulating $Y$ will change the value of $X$. By the same logic, $X$ is not the cause of $Y$ because manipulating $X$ will not change $Y$.

When history is infinite, GC works as expected

$$\mathbb{E}[\mathbf{Y}_t|\mathbf{X}^t, \mathbf{Y}^t] = \mathbb{E}[\mathbf{Y}_t|\mathbf{Y}^t] = \mathbf{Y}_t$$
$$\mathsf{MSE}(\mathbf{Y}_t, \mathbb{E}[\mathbf{Y}_t|\mathbf{Y}^t]) = \mathsf{MSE}(\mathbf{Y}_t, \mathbb{E}[\mathbf{Y}_t|\mathbf{X}^t, \mathbf{Y}^t]) = 0$$
$$\Rightarrow X \text{ does not cause } Y \quad (\text{correct})$$

However, when only 1 past observation is given (finite history), GC reaches the wrong conclusion

$$\mathsf{MSE}(\mathbf{Y}_t, \mathbb{E}[\mathbf{Y}_t|\mathbf{Y}_{t-1}]) \geq \mathsf{MSE}(\mathbf{Y}_t, \mathbb{E}[\mathbf{Y}_t|c\mathbf{Y}_{t-2}, \mathbf{Y}_{t-1}]) \geq \mathsf{MSE}(\mathbf{Y}_t, \mathbb{E}[\mathbf{Y}_t|\mathbf{X}_{t-1}, \mathbf{Y}_{t-1}])$$
$$\Rightarrow X \text{ causes } Y \quad (\text{incorrect})$$

Thus, GC may detect edges in both direction ($X \to Y$ and $Y \to X$) for a pair of variables when limited history is given. It can be shown in a similar fashion that if $X$ causes $Y$ and $Y$ causes $Z$ ($X$ is the indirect cause of $Z$), $Y$ will not be able to shield $Z$ from $X$ if only limited history is given. Thus, GC will also detect edges for indirect causes in both direction ($X \to Z$ and $Z \to X$).

In GLACIAL, we implement two additional post-processing steps to remove these false positives. Let $S(X|Y)$ be the statistic (e.g. the t-test) that tests for the positivity of $\Delta\mathsf{MSE}$ from several train/test splits. Thus $S(X|Y)$ can be viewed as a test for whether $Y$ causes $X$.

**1. Orient bidirectional edge**  If $S(X|Y) < S(Y|X)$ remove $Y \to X$, else remove $X \to Y$. This step is similar to prior work such as (Hoyer et al., 2008; Zhang & Hyvärinen, 2009a; Janzing et al., 2012; Kocaoglu et al., 2017) which leverages causal asymmetry to determine the causal direction (the direction with the bigger effect is regarded as the causal direction). T-statistic has been shown to be informative for causal discovery (Weichwald et al., 2020). Appendix H presents a mathematical justification for this heuristic.

**2. Remove indirect edge**  Remove edge $X \to Y$ if there exists an alternative path $(X := U_0, U_1, \dots, Y := U_k)$ from $X$ to $Y$ or a path $(Y := U_0, U_1, \dots, X := U_k)$ from $Y$ to $X$ such that

$$S(Y|X) < \min(\{S(U_{j+1}|U_j); \ j \in 0, \dots, k-1\})$$

Intuitively, if there is an alternative path on which the effect of the weakest edge is greater than the effect of $X \to Y$ then $X$ is likely an indirect cause of $Y$. A complete description of GLACIAL is shown in Algorithm 1. Ablation in Section 5.2 shows the contribution of these two post-processing steps.

### 3.5 Runtime Complexity

Since GLACIAL uses a single multi-task RNN to check relationships between all variable pairs, the number of RNN models trained by GLACIAL is independent of the number of variables and is equal to the number of data splits (see Algorithm 1 line 1-2). For example, with 4 repetitions of 5-fold cross-validation, GLACIAL needs to train 20 different RNN models. This number is the same whether there are 10 or 100 variables. Obviously, having more variables will lead to longer execution time per batch but much of the computation is parallelizable (as long as the batch fits into GPU memory). Therefore, the runtime complexity is mostly dominated by the number of RNN models that must be trained.

# 4 Experimental Set-up

In addition to the problems listed in Section 3, CD methods often struggle when (1) relationships are non-linear, (2) the number of variables is large, or (3) a node has many parents. The subsequent experiments are designed to show GLACIAL's efficacy and to show that GLACIAL is less affected by these problems. First, the simulations in Section 4.2 include both non-linear trajectories and linear random-walk trajectories. GLACIAL is also applied on real multivariate data (Section 4.3) which most likely include non-linear trajectories. Second, there is a simulation with a moderate-size graph consisting of 39 nodes to demonstrate scalability. Third, the simulation with the 39-node graph includes one node (i.e. 22) with 18 direct causes.

## 4.1 Baselines

We benchmark GLACIAL against both CD methods for cross-sectional data and CD methods for timeseries. Section 5.1 shows only representative and competitive baselines (see Appendix A for the remaining baselines).

**CD Methods for Cross-Sectional Data**   We compare against PC, FCI (Spirtes et al., 2000), GFCI (Ogarrio et al., 2016), and *Sort-N-Regress* (Reisach et al., 2021) (SnR). GFCI combines GES and FCI into a single algorithm. SnR is a simple baseline to ensure that benchmarked approaches go beyond exploiting differences in variables' marginal variance (Reisach et al., 2021). As these approaches assume independent observations, only first timepoints (observations) of individuals are used. In most longitudinal studies, individuals are guaranteed to have first timepoints (but not other timepoints). Hence, using the first timepoints will result in the most number of independent timepoints with the least amount of missing data in real-world datasets. Besides, using all timepoints led to worse performance in our preliminary experiments using simulated data. Similar to Shen et al. (2020), GFCI is run multiple times (i.e. 20) using different bootstraps of individuals' first timepoints, resulting in multiple graphs. Only edges appearing in more than half of the resultant graphs are kept in final graph. Using a higher threshold (80%) led to worse result (see Appendix F).

**CD Methods for Timeseries Data**   We also adopt SVAR-GFCI (Malinsky & Spirtes, 2019), PCMCI+ (Runge, 2020), DYNOTEARS (Pamfil et al., 2020), and several GC-based approaches as baselines. GC-based approaches include linear GC and more recent neural GC tests: cMLP and cLSTM from (Tank et al., 2021), TCDF from (Nauta et al., 2019), SRU and eSRU from (Khanna & Tan, 2020). For linear GC[1], F-statistic was used to test for presence of edges using the same threshold as in GLACIAL. For longitudinal data, one could either (1) estimate one causal graph for each individual and aggregate the graphs or (2) estimate just one graph using concatenated data from all the individuals. Since (1) often fails when the number of timepoints per individual is sparse, (2) was used instead. Causal discovery using concatenated individuals' data has been investigated in (Di et al., 2019; Qing et al., 2021). Besides, linear GC could output false positives when timeseries are non-stationary (He & Maekawa, 2001). One could make the timeseries stationary by calculating the difference or the log difference between timepoints (Stock & Watson, 2012). However, using differences led to worse results so we report the results using the original timeseries instead.

The input to SVAR-GFCI[2] and PCMCI+[3] are also the concatenated timeseries from all the individuals. For DYNOTEARS[4] which can accept timeseries from multiple individuals, the timeseries are not concatenated. The hyper-parameters of SVAR-GFCI, PCMCI+, DYNOTEARS and neural GC tests are selected based on the suggestions in their original publications.

**Missing data**   For data with missingness, TDPC (Strobl et al., 2018) and MVPC (Tu et al., 2019) are used instead of GFCI. For a dataset, each algorithm is run 20 times and the results are aggregated using the same 50% threshold. As far as we know, there is no prior work on applying causal discovery methods to timeseries data with missing values. Therefore, we used linear interpolation to fill out missing values in the data before applying these methods (linear/neural GC, SVAR-GFCI, PCMCI+, and DYNOTEARS). It

---

[1]https://github.com/statsmodels/statsmodels
[2]https://github.com/cmu-phil/tetrad
[3]https://github.com/jakobrunge/tigramite/
[4]https://github.com/quantumblacklabs/causalnex

may not be feasible to apply more complex interpolation methods due to the limited number of timepoints (especially after discounting missing values).

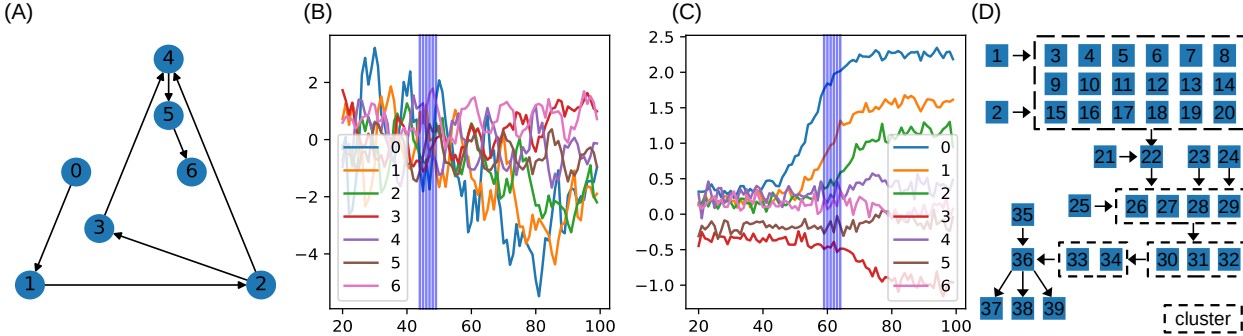

Figure 2: *Simulation. (A) 7-node graph having all basic structures (chain, fork, collider). (B) Individual with random-walk trajectories (data before standardizing to zero mean and unit variance). Only timepoints under vertical lines are observed. (C) Individual with sigmoid trajectories. (D) More realistic 39-node graph resembling the RTK/RAS signaling pathway. Nodes in the same cluster have the same causal relations.*

## 4.2   Simulated Data

The sample size in the simulations was set to 2000 individuals, roughly the size of the ADNI dataset (see Section 4.3). Only six timepoints are extracted from each individual's timeseries to simulate sparse observations (see Appendix D for results with 24 timepoints). We consider two scenarios. First, the temporal dynamics are parameterized via the sigmoid function, which is a widely used model for the trajectories of biomarkers, e.g., in Alzheimer's disease (Jack Jr et al., 2013). In the second scenario, we implement random-walk series. See Appendix C for further details. As causal structure of simulated data may leak through variables' marginal variance, the data are standardized to zero-mean and unit-variance to prevent CD algorithms from gaming the simulated data (Reisach et al., 2021).

Figure 2 shows the causal graphs used for generating the synthetic data. The first graph (7 nodes) contains all the basic structures, namely chain, fork, and collider. The second graph (39 nodes) is used to demonstrate GLACIAL's scalability. This graph is inspired by the RTK/RAS signaling pathway in oncology and is taken from (Sanchez-Vega et al., 2018). The second graph is a realistic target that a causal discovery algorithm should be able to find from observational data. Since the shape of the evolution of signaling proteins is not known, we use Gaussian random-walk as the sample path function. To simulate missingness (completely at random; MCAR), the values for each timeseries of an individual are independently dropped at fixed rate $p \in \{0.1, 0.3, 0.5\}$. Since real data missingness may be more adverse than MCAR, results on simulated missing data are optimistic estimates of performance. The missingness rate is chosen to match the rate in real data (see next Section). Since values from different timeseries are dropped independently, the resulted data could contain individuals with all timepoints having at least one missing values.

## 4.3   Real-world Data from an Alzheimer's Disease Study

We use ADNI (Jack Jr et al., 2008), a longitudinal study of Alzheimer's disease (AD) and consists of 1789 individuals. Each individual in ADNI has about 7 timepoints on average. The ADNI study tracks multiple AD biomarkers such as region-of-interest (ROI) volumes (e.g. hippocampal) derived from structural MRI scans, cognitive tests (e.g. ADAS13), proteins (e.g. amyloid beta) derived from cerebral spinal fluid samples, and molecular imaging that captures the brain's metabolism (e.g. FDG PET). The missingness rates vary for different biomarkers, ranging from 30% (ADAS13) to around 80% (FDG PET). The variables used are shown in Figure 7c and described in detail in Appendix G.

### 4.4 Metrics

F1-score, which is the harmonic mean of precision and recall, is used to quantify different approaches' performance. Note that we assume that there is a ground-truth (directed) graph that describes causal relations. Each method will also return a list of directed edges between variables. Precision is the ratio of correctly identified edges over all predicted edges, while recall is the ratio of correct edges over all ground-truth edges. A predicted edge is considered incorrect if the edge does not exist in the ground-truth graph or the predicted direction contradicts the ground-truth direction. Thus, a predicted bidirectional edge would be incorrect if the ground-truth edge has only one direction.

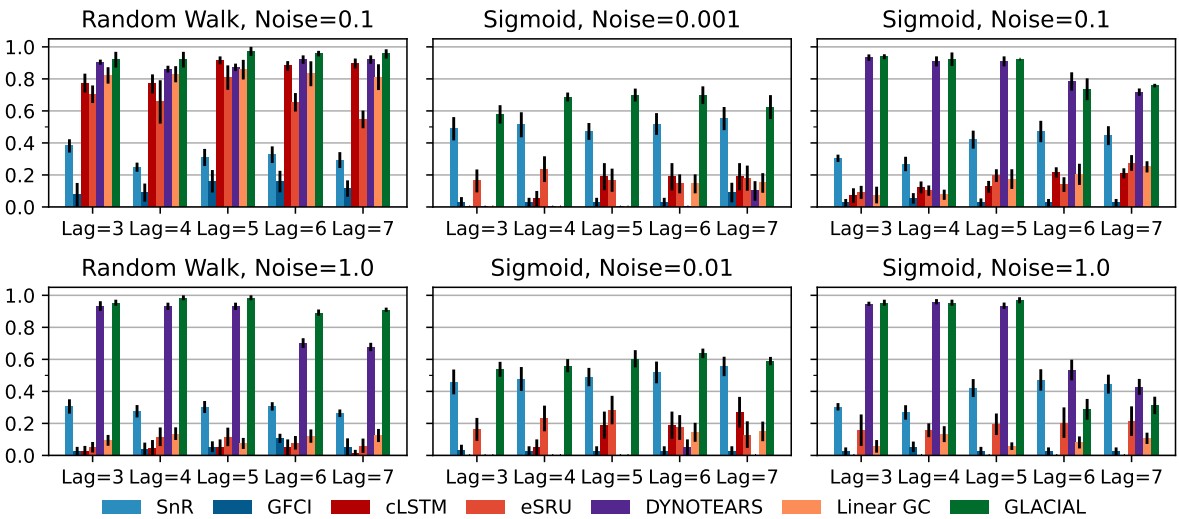

Figure 3: *Average F1-scores at different settings of sample path, lag-time and measurement noise (7-node graph). GLACIAL outperforms baselines in most settings (see Appendix A for more comparisons).*

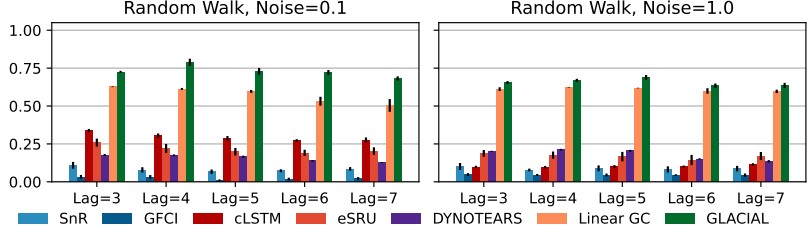

Figure 4: *Average F1-scores at different settings of lag-time and measurement noise (39-node graph, Gaussian random-walk). GLACIAL outperforms baselines in most settings (see Appendix A for more comparisons).*

## 5 Experimental Results

### 5.1 Simulated Data

**7-node graph** For random-walk data, GLACIAL outperforms the baselines for various lag-times and measurement noise levels (Figure 3, 1st column). Similarly, GLACIAL also outperforms the baselines, for the sigmoid data (2nd and 3rd column). GLACIAL's performance dips (3rd column) when input history (5 years) is shorter than the lag-time (6 or 7 years). This dip is more pronounced when measurement noise is high (3rd column, bottom).

DYNOTEARS fails to detect causal relations in systems with almost deterministic dynamics (2nd column) even though it is the best baseline. System with deterministic dynamics is also challenging for linear GC (Peters et al., 2017) although it is slightly better than DYNOTEARS (F1-score < 0.2). Interestingly, GLACIAL still works in these systems (F1-score = 0.6). Only GLACIAL manages to consistently beat the strong SnR (*Sort-N-Regress*) baseline, demonstrating the proposed method's strength.

**39-node graph**   Even though DYNOTEARS is the best baseline for the 7-node graph, its performance on the big graph is worse than linear Granger (Figure 4). GLACIAL consistently outperforms all baselines on this big graph when the sample path is Gaussian random-walk. GLACIAL performs quite well despite the presence of a cluster of direct causes whose contribution to the node "22" may be too small to be detected.

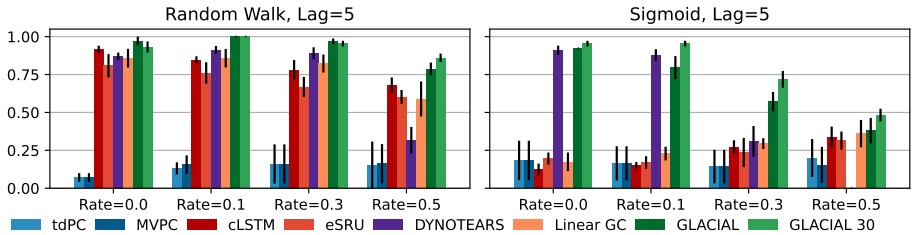

Figure 5: *Average F1-scores at various levels of missing at random. Lag-time=5. Noise level = 0.1. GLACIAL usually outperforms baselines. Running GLACIAL for more repetitions (i.e. 30 instead of 4, denoted as GLACIAL 30; see Section 3.3) can improve performance when dealing with missing data.*

**Missing data**   Figure 5 shows F1-scores at different degrees of missingness. GLACIAL outperforms TDPC and MVPC, CD approaches tailored for missing data, by better exploiting the temporal dynamics within individuals' timeseries. GLACIAL also outperforms CD methods for timeseries such as cLSTM and DYNOTEARS. Although being the best baseline, DYNOTEARS often fails when the missingness level is high (>0.3). When half of the values are missing (=0.5), GLACIAL can still infer some causal relations. As an aside, GLACIAL's performance on missing data can be improved with more repetitions (see Appendix E).

## 5.2   GLACIAL's Post-processing Ablation

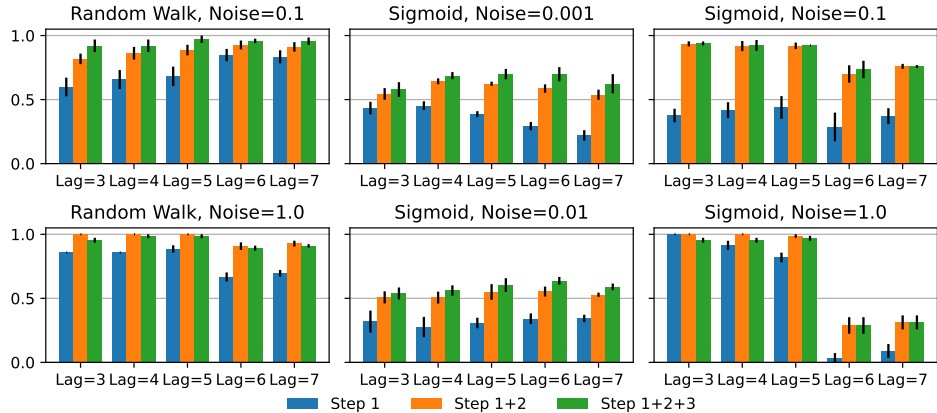

Figure 6: *Contribution from GLACIAL's heuristics to F1-scores (7-node graph simulation).*

GLACIAL's first step tests for edges in the causal graph by comparing the difference in MSE on hold-out individuals. However, when test individuals are only sparsely observed for a limited number of times, this step may find spurious edges (edges from effect to cause or edges between indirect cause, e.g. a grand-parent,

Table 1: *GLACIAL's results vary little with different hyper-parameters. Lag-time=5. Noise level = 0.1. L: number of layers, D: size of hidden layer, R: learning rate (R1: 1E-3, R2: 3E-4, R3: 1E-4)*

| Simulation | **L1-D256-R2** | L1-D128-R2 | L1-D512-R2 | L2-D256-R2 | L1-D256-R1 | L1-D256-R3 |
|---|---|---|---|---|---|---|
| **Random-walk** | 0.97±0.06 | 0.96±0.06 | 0.96±0.06 | 0.96±0.06 | 0.92±0.09 | 0.97±0.06 |
| **Sigmoid** | 0.92±0.00 | 0.92±0.09 | 0.91±0.08 | 0.94±0.03 | 0.92±0.00 | 0.92±0.00 |

and effect). To address this problem, GLACIAL has two additional heuristics: one (Step 2) to remove edges from effect to cause and another (Step 3) to prune edges between indirect cause and effect. Figure 6 shows the contribution of these two post-processing heuristics to F1-scores at various lag-times and noise levels (7-node graph simulation). The first heuristic (Step 2) consistently leads to better results. While the second heuristic (Step 3) is beneficial most of the time, it can sometime result in performance degradation. Thus, when applying GLACIAL to real data, it is recommended to compare the outputs with and without the second heuristic to decide which output is more plausible.

### 5.3 GLACIAL's Hyper-parameter Sensitivity Ablation

Since GLACIAL uses neural network for inference, ones may think that its results are sensitive to the choice of hyper-parameters. We analyzed GLACIAL's performance as the hyper-parameters vary. Table 1 shows the performance of GLACIAL while varying (1) the number of hidden layers, (2) the size of the hidden layer(s), and (3) the learning rate used. GLACIAL's results seem quite robust to the choice of hyper-parameters.

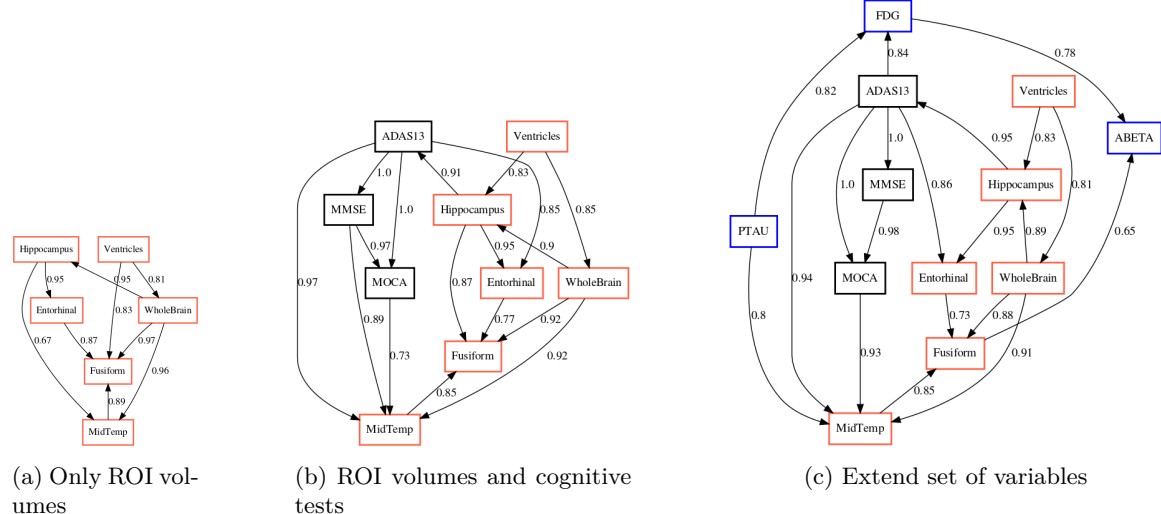

(a) Only ROI volumes

(b) ROI volumes and cognitive tests

(c) Extend set of variables

Figure 7: *GLACIAL's predicted interaction of ADNI biomarkers. ROI volumes are in red, cognitive tests are in black, and the rest are in blue. ABETA: amyloid beta, PTAU: phosphorylated tau. Edge weights are frequencies at which edges were detected in multiple runs.*

### 5.4 Results on ADNI Data

The output of applying GLACIAL to different sets of ADNI biomarkers are shown in Figure 7. Edge weights denote the frequencies at which edges were detected in multiple runs. Most of the edges are consistently detected across different runs with the exception of "Hippocampus → MidTemp" (67%, Figure 7a) and "Fusiform → ABETA" (65%, Figure 7c). Although GLACIAL's neural model assumes MCAR and missingness in ADNI data may be more adverse than that, GLACIAL' result seems promising. There is a high degree of agreement between the 3 graphs which all show the "Ventricle" is a source in the causal graph and "Fusiform" is at the end of the chain. The presence of the edge "Hippocampus → Entorhinal" is also consistent with literature. In comparison, baselines' outputs are less interpretable (Figure 8; more results

are in Appendix G). The outputs of DYNOTEARS and linear Granger contain hardly any edge between ROI volumes while the outputs of cLSTM and eSRU have bidirectional edges.

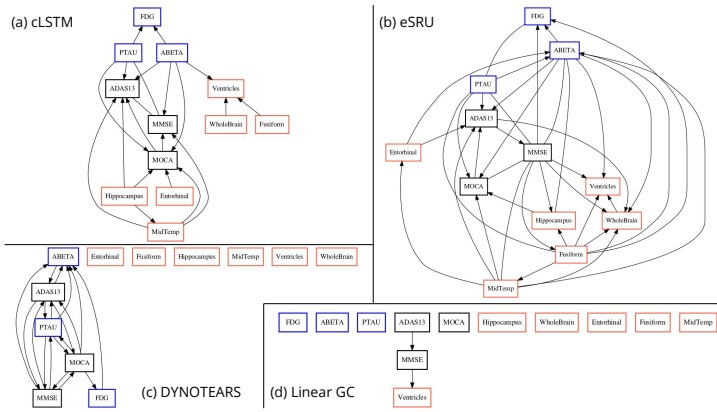

Figure 8: *Baseline approaches' predicted interaction of ADNI biomarkers*

## 6 Discussion

Longitudinal studies, in which multiple individuals are sparsely observed for a limited number of times, are particularly common in population health applications. Longitudinal studies often track many variables, which are likely governed by nonlinear dynamics that might have individual-specific idiosyncrasies. Yet, longitudinal studies are not amenable to the popular Granger causality (GC) analysis, since GC was developed to analyze a single multivariate densely sampled timeseries. Furthermore, real-world longitudinal data often suffer from widespread missingness. We developed GLACIAL which combines the GC framework with a machine learning based prediction model to address this need. GLACIAL treats individuals as independent samples and uses average prediction accuracy on hold-out individuals to test for causal relations.

GLACIAL exploits a single multi-task neural network trained with input feature dropout to efficiently probe links. GLACIAL places no restriction on the design of the neural network predictor. This flexibility allows future extensions of our work. For example, Transformers (Vaswani et al., 2017) or Neural ODEs (Chen et al., 2018) can be used instead of the RNN architecture.

**Limitations and Future Work**   Although we showed GLACIAL working well in many settings (varying lag-times, noise levels, and missingness degree), there are some questions remained that need further investigation. We focused on real-valued variables since they are the most common but extending GLACIAL to discrete variables by adopting techniques in (Peters et al., 2010; Cai et al., 2018; Huang et al., 2018) would make GLACIAL analysis applicable to more longitudinal studies. Furthermore, studying GLACIAL's behaviors under missingness other than MCAR is important despite GLACIAL outputting plausible graphs on possibly non-MCAR data (ADNI). Besides, GLACIAL assumes that there is no feedback, hidden confounder, or instantaneous effect. Thus, before applying GLACIAL, it is critical to verify whether these assumptions hold to ensure that causal relations inferred by GLACIAL are valid. The third assumption in particular requires that the sampling resolution is high enough to capture transient changes (e.g. impulses) or temporal orderings between causal pairs with short time lags. Although causal discovery when some assumptions are violated has been studied in the past (for example, presence of hidden confounders Spirtes et al. (2000) or presence of instantaneous effects (Danks & Plis, 2013; Gong et al., 2015; 2017)), extending these techniques to longitudinal studies is still an open question. We leave these questions for future work.

**Broader Impact Statement**

Since GLACIAL is based on Granger causality framework, GLACIAL also assumes that there is no hidden confounders and no instantaneous effects. Thus, causal relations inferred by GLACIAL should be interpreted with caution, especially when these assumptions do not hold, to avoid misleading conclusions.

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
