# OpenReview forum: "GLACIAL: Granger and Learning-based Causality Analysis for Longitudinal Studies"
_TMLR — Rejected by TMLR_

### Review · Reviewer_N2LN · 2023-02-14

**Summary Of Contributions:**

The problem setup is: there is the time series data of many individuals, where for each individual, there are time series of identical set of variables, but some values in some time series may be missing. The authors aim to discover the causal relations among the involved variables. They resort to granger causality. More specifically, for any two variables  $X$ and $Y$, they train two RNNs to estimate $Y_t$ with all the historical variables and estimate $Y_t$ with all historical variables except for $X_0,\cdots, X_{t-1}$, respectively. If there is not a significant difference between the MSEs for the two models, then $X$ does not cause $Y$; while $X$ causes $Y$ otherwise. By considering all variable pairs, the whole causal relations are learned. In light of the possibility that the false causal relations $X\rightarrow Y$ could be learned when $X$ only has an indirect causal relation on $Y$, the authors propose two heuristic rules to avoid it.

**Audience:**

Yes

**Broader Impact Concerns:**

No.

**Claims And Evidence:**

Yes

**Requested Changes:**

1. It is expected to have more novel insights about the problem tackled.
2. There are some weird sentences. E.g., "These methods would use one observation per individual and thus is not designed to detect causal relations reflected in temporal dynamics. **Hence**, we believe there is a lack of methods for causal discovery in longitudinal studies that consist of multiple individuals with sparse observations. "
3. There are some confusing parts. E.g., "when systems have a lot of variables, some variables may have many direct causes. In this case, regression-based GC (Granger, 1969; Lütkepohl, 2005) often fails to detect some of the true causal relations". It is better to give an explanation about the reason. In addition, according to my understanding, the method in this paper is also regression-based.
4. The second step (remove indirect edge) in Sec. 3.3 seems not sensible. It will be better if the authors provide more illustrations( or make some modifications).
5. The computation cost needs to be discussed.
6. no latent confound -> no latent confounder.
7. It will be better if the authors can clarify whether the disadvantages of related works (in the last paragraph of Page 1) are addressed by their proposed method.

**Strengths And Weaknesses:**

Advantage:
1. The problem tackled in this paper is interesting.
2. The method is useful in practice.

Disadvantage:
1. The contribution/novelty is limited. I did not find new insights about the problem the authors want to tackle.  Indeed, the causal discovery of time series with missing values is worthy to study, but the proposed method seems to be a little well-known.

2. The computation cost is too large. If there are $d$ variables, there are $2^d$ variables pairs, thus there are $2^{d+1}$ trained RNN models.

3. The second step (remove indirect edge) in Sec. 3.3 seems not sensible. Consider an example $Z_{t+1}=X_t + \epsilon_1, Y_{t+2}=0.5 * X_t - Z_{t+1} + \epsilon_2$. Ideally(although I do not think p-value is a quite good statistic), there is $S(Y|X)<min(S(Y|Z), S(Z|X))$, but there should be $X\rightarrow Y$ in the summary graph.

---

> ### Author Response · Authors · 2023-03-25
> **Response to Reviewer N2LN**
>
> **I did not find new insights about the problem ... causal discovery of time series with missing values is worthy to study, but the proposed method seems to be a little well-known.**
>
> We would like to emphasize that the problem we are considering is longitudinal data not timeseries data. Longitudinal data is very distinct from regular timeseries data. Critically, the problem we consider includes multiple subjects (or systems), sparsely sampled at different time points.
> Although the Granger framework is well-known, to the best of our knowledge, all prior work based on Granger only considered timeseries data (i.e., a single subject/system, densely sampled over time). Thus, when applying approaches proposed for timeseries data or cross-sectional data to real longitudinal data (ADNI dataset), their results are inadequate, most likely because the methods are inappropriate. Our approach is the first to specifically deal with longitudinal data (with multiple subjects, and a few timepoints per subject).
> We have moved the ADNI results of baseline approaches to the main text to show their insufficiency and thus emphasizing a need for a method that specifically deals with longitudinal data.
>
> **If there are d variables, there are 2^d variables pairs, thus there are 2^{d+1} trained RNN models.**
>
> We are sorry for not being clear. For a given dataset, we only train a single RNN which makes our approach computationally very efficient. We train several RNNs because we are considering multiple data splits. That is, the number of RNNs we obtain is equal to the number of data splits (Algorithm 1, line 1-2). Thus, with 4 repetitions of 5-fold cross-validation, there will be 20 data splits and hence 20 RNNs to be trained. The number of RNNs does not depend on the number of variables. If this number was 2^{d+1} (it is not), getting the results for the 39-variable problems (Section 5.1) would require training 2^40 (~1 trillion) RNNs. This would certainly be beyond our computational budget.
> We have added this discussion to Section 3.5.
>
> **The second step (remove indirect edge) in Sec. 3.3 seems not sensible. Consider an example... Ideally (although I do not think p-value is a quite good statistic), there is S(Y|X) < min(S(Y|Z), S(Z|X)), but there should be X→ Y in the summary graph.**
>
> By statistic, we mean the t-statistic of the t-test not the p-value. Since S is the t-statistic of the difference in MSE errors and the data would be standardized to unit-variance, it is unclear what value S takes analytically. Thus, it’s hard to say with certainty that S(Y|X) < min(S(Y|Z), S(Z|X)).
> We tried to simulate this example. In 30 simulations (2k subjects each), X → Y is removed by the heuristic in 5 out of 10 simulations (50%) when this edge is detected. In the remaining 20 simulations, this edge is not even detected in the first stage so the failure could also be due to errors in MSE estimation. We acknowledge that this heuristic is not perfect and our ablation also shows this in some cases. Thus, we have moved the ablation into the main text and updated the writing to acknowledge this point (Section 5.2).
>
> **It will be better if the authors can clarify whether the disadvantages of related works are addressed by their proposed method.**
>
> We had indeed not explicitly explained how GLACIAL addresses these disadvantages. The last paragraph of Page 1 listed 3 disadvantages: 1) may produce wrong graphs when relationships are non-linear, 2) may not scale to large graphs, 3) may not detect parents of a node with many parents. We showed via experiments that GLACIAL is less affected by these problems.
>
> 1\) GLACIAL outperformed baselines on simulations using non-linear trajectories and produced more promising outputs for ADNI data which most likely have non-linear trajectories. 2) our simulation includes a moderate-size graph with 39 nodes. 3) GLACIAL performance gap over baseline approaches on the 39-node graph simulations supported this point as there is one node (i.e. 22) with 18 direct causes.
> We have added this discussion at the beginning of Section 4 in the revised article.
>
> **...some confusing parts... the method in this paper is also regression-based.**
>
> When a system has a lot of variables, some may have many direct causes and the coupling between a variable to some of its parents may be weak. As the detection power of regression-based GC diminishes with increasing number of variables, regression-based GC may fail to detect the weak couplings between a node and its parents. We have updated the draft to try to convey the reason.
> By regression-based, we refer to the standard approach of fitting a linear regression model and interpreting its parameters. On the contrary, the GLACIAL test doesn’t involve interpreting the parameters but instead analyzes the predictive performance on hold-out subjects. Hence, it’s more prediction-based rather than regression-based.
>
> **...some weird sentences.**
>
> We have revised the text to reflect this feedback.

---

### Review · Reviewer_TzJM · 2023-03-04

**Summary Of Contributions:**

This paper aims to learn causal relations under the setting of longitudinal study design. Specifically, the paper proposed a method, named GLACIAL, to discover causal relations between multiple variables in a longitudinal study by combining Granger causality (GC) with a multi-task neural model (a practical machine-learning-based approach). The experimental results on synthetic and real-world data show that the proposed method outperforms baselines.

**Audience:**

Yes

**Claims And Evidence:**

Yes

**Requested Changes:**


It would be better to give a more detailed mathematical definition of the setting of longitudinal studies.

This paper claims that GLACIAL is able to solve the causal discovery problem in longitudinal studies, but it does not clearly explain how GLACIAL solves it. The paper says that this problem will be discussed in Section 3.1. However, Section 3.1 only gives the definition of GC MSE formulation, it might need to further discuss how GLACIAL solve it.

Section 3.2 mentions that the ability to handle missing values comes from the RNN model proposed by Nguyen et al., it might need to further disclose how to incorporate the model into the GLACIAL method.


**Strengths And Weaknesses:**

Pros:
1. This paper investigates learning causal relationship problems from longitudinal data. This is an important and challenging task for the related community.
2. The proposed method, GLACIAL, is able to detect the causal relations between multiple variables in a longitudinal study.
3. The experimental results verify the effectiveness of the proposed methods.

Cons:
1. The problem is not properly stated. I think that the problem statement (i.e., the longitudinal studies), should be provided before Section 3.
2. There is no complexity analysis and theoretical guarantee for the proposed algorithm.
3. The description of this model is only shown in Figure 1. It would be better if there are more discussions about the model.

---

> ### Author Response · Authors · 2023-03-25
> **Response to Reviewer TzJM**
>
> **The problem is not properly stated. I think that the problem statement (i.e., the longitudinal studies), should be provided before Section 3.
> It would be better to give a more detailed mathematical definition of the setting of longitudinal studies.**
>
> We agree that explaining the longitudinal setup in more detail and upfront would be helpful to the readers. We have added the definition of the longitudinal setup in Section 3.1.
>
> **There is no complexity analysis and theoretical guarantee for the proposed algorithm.**
>
> We have added a computational complexity analysis to Section 3.5. In summary, the number of models that need to be trained is independent of the number of features and is linear in the number of cross-validation folds. Hence, GLACIAL can be applied to problems with a large number of variables. GLACIAL is based on Granger’s framework so GLACIAL inherits all the theoretical guarantees that Granger causality has. We are not sure what other theoretical guarantees are expected of causal discovery algorithms and would be grateful for any further suggestion.
>
> **The description of this model is only shown in Figure 1. It would be better if there are more discussions about the model.
> Section 3.2 mentions that the ability to handle missing values comes from the RNN model proposed by Nguyen et al., it might need to further disclose how to incorporate the model into the GLACIAL method.**
>
> We thank the reviewer for these suggestions. We have added more discussion about the model and its ability to handle missing data in Section 3.3
>
> **This paper claims that GLACIAL is able to solve the causal discovery problem in longitudinal studies, but it does not clearly explain how GLACIAL solves it. The paper says that this problem will be discussed in Section 3.1. However, Section 3.1 only gives the definition of GC MSE formulation; it might need to further discuss how GLACIAL solves it.**
>
> We thank the reviewer for the feedback. We have added more discussion in this section to further show how GLACIAL solves causal discovery in the longitudinal setting.

---

### Review · Reviewer_yopp · 2023-03-12

**Summary Of Contributions:**

The authors propose GLACIAL, a Granger causality (GC) method suitable for dealing with longitudinal data which is sparsely observed and may suffer from missingness. The proposed method combines the ideas of GC with a multi-task neural model. The features of the proposed approach are as follows:

1. Treats individuals as independent samples and uses MSE to determine causation
2. Uses feature dropout to handle large number of variables
3. Uses interpolation to handle missing values
4. Uses post-processing heuristics to distinguish between direct and indirect causes

Experiment results on synthetic and real data is provided to show the performance of the method.

**Audience:**

Yes

**Broader Impact Concerns:**

As mentioned above, the proposed method has two very strong assumptions: no hidden confounders, no instantaneous effects. Without these two assumptions GC may not have any causal interpretation. These two assumptions are violated in most of the complex real-world setting. Especially the second assumption is violated whenever the sampling resolution is low. Therefore, the use of the proposed method and interpreting the association-based results as causal can easily lead to misleading conclusions which is of great concern in, e.g., public health applications. Same type of concern holds for the proposed post-processing heuristics which again can lead to misleading conclusions.

**Claims And Evidence:**

No

**Requested Changes:**

- The authors should clarify that treating individuals as independent samples is not a novel point of view and very common in longitudinal data analysis [Hernan & Robins, Causal inference, part III]
- It seems important that the authors formalize the mathematical conditions under which the proposed post-processing heuristics in fact make the right decision
- The authors should clarify the limitations of GC in terms of actual causal interpretation. Specifically, the fact that GC is essentially a correlation-based method and only under strong conditions can have a causal interpretation. Providing such an explanation is crucial for readers unfamiliar with the causality literature
- The authors should clarify if/how different approaches for hyper parameter tuning affect the results
- It seems important that the authors provide more detailed explanations about how the missingness in the data is handled. Assuming that we have missing at random in the data, how is the adjustment step done?

**Strengths And Weaknesses:**

Strengths:
- The paper is well-written and easy to follow
- The authors pay special attention to longitudinal data with missing values and propose an implementation of GC for this setting, which can be of interest to many practitioners
- The proposed method pays attention to the challenge of missing values and having finite history

Weaknesses:
- The proposed method has two very strong assumptions: no hidden confounders, no instantaneous effects. Without these two assumptions GC may not have any causal interpretation
- The proposed post-processing heuristics does not have any mathematical basis. It seems possible to come up with examples in which the heuristic makes the wrong decision
- Due to the fact that neural networks are used in the method, there are questions regarding the interpretability of the results. Specifically, how sensitive the results are to the choice of hyper parameters
- The authors interpret faithfulness assumption as "statistical dependency implies causal relation". This is not the precise definition of faithfulness

---

> ### Author Response · Authors · 2023-03-25
> **Response to Reviewer yopp**
>
> **The proposed method has two very strong assumptions: no hidden confounders, no instantaneous effects. Without these two assumptions GC may not have any causal interpretation.
> These two assumptions are violated in most of the complex real-world setting. Especially the second assumption is violated whenever the sampling resolution is low. Therefore, the use of the proposed method and interpreting the association-based results as causal can easily lead to misleading conclusions which is of great concern in, e.g., public health applications.
> The authors should clarify the limitations of GC in terms of actual causal interpretation. Specifically, the fact that GC is essentially a correlation-based method and only under strong conditions can have a causal interpretation. Providing such an explanation is crucial for readers unfamiliar with the causality literature**
>
> We agree with the reviewer about the need to discuss the assumptions behind Granger causality. It is true that any causal claim discovered should be interpreted with caution because the assumptions may be invalid and people who are not machine learning experts may not be aware of these assumptions. We have updated Section 6 (Discussion) and added the Broader Impact Statement to address this concern.
>
> **The proposed post-processing heuristics does not have any mathematical basis. It seems possible to come up with examples in which the heuristic makes the wrong decision.
> It seems important that the authors formalize the mathematical conditions under which the proposed post-processing heuristics in fact make the right decision
> Same type of concern holds for the proposed post-processing heuristics which again can lead to misleading conclusions.**
>
> We acknowledge that the previous version of our paper did not provide any mathematical justification for the heuristics. We have added some mathematical reasoning for the first heuristic in Appendix H. This is only for a simple case when the data generation is a second-order linear system and the predictors are given only a single timepoint to predict the next timepoint. We show that the first heuristic will remove the falsely detected edge. Whether there exists a proof for the general case of the first heuristic and a proof for the second heuristic is still an open question and will be left for future work.
>
> **Due to the fact that neural networks are used in the method, there are questions regarding the interpretability of the results. Specifically, how sensitive the results are to the choice of hyper parameters
> The authors should clarify if/how different approaches for hyper parameter tuning affect the results**
>
> We thank the reviewer for these pointers. We have conducted additional ablation experiments to vary (1) the number of hidden layers in the RNN, (2) the size of the hidden layers in the RNN, and (3) the learning rate used to train the RNN. The results of these additional experiments are shown in Section 5.3 of the latest version of our paper. In short, our experiments demonstrate that GLACIAL is not very sensitive to the choice of hyper parameters. That said, we would like to emphasize that any causal discovery method, including GLACIAL, depends on modeling assumptions and most prior work that relies on the Granger approach employ simple linear models.
>
>
> **The authors should clarify that treating individuals as independent samples is not a novel point of view and very common in longitudinal data analysis [Hernan & Robins, Causal inference, part III]
> The authors interpret faithfulness assumption as "statistical dependency implies causal relation". This is not the precise definition of faithfulness**
>
> We thank the reviewer for pointing this out. We now cite to the Hernan & Robins paper. We have also modified the text and now avoid referring to the faithfulness assumption.
>
>
> **It seems important that the authors provide more detailed explanations about how the missingness in the data is handled. Assuming that we have missing at random in the data, how is the adjustment step done?**
>
> We have added this information in the revised Section 3.3.

---

### Decision · Action_Editors · 2023-04-14

**Recommendation:** Reject

**Comment:**

The authors propose a new Granger causality method called GLACIAL that can handle longitudinal data with missing values. The method uses a multi-task neural model, feature dropout, and interpolation to handle the challenges of sparsity and missingness. Post-processing heuristics are used to distinguish between direct and indirect causes. The paper provides experimental results on synthetic and real data to show the method's performance.

Strengths of the paper include its well-written and easy-to-follow style, the new setup on longitudinal data with missing values. However, this paper has several limitations. First, the assumptions under which one can interpret the correlation-based approach as causal relations are not clear. Second, the post-processing rely on heuristics lack scientific evidence. Third, the treatment of randomness in missing data is not rigorous. Based on the current review comments, I do not recommend acceptance of this paper.

**Audience:**

Yes

**Claims And Evidence:**

No.

The post-processing heuristics are not supported by scientific evidence.